# Associations between physical activity prior to infection and COVID-19 disease severity and symptoms: results from the prospective Predi-COVID cohort study

Laurent Malisoux  ,[1] Anne Backes,[1] Aurélie Fischer,[1] Gloria Aguayo,[1] Markus Ollert,[2,3] Guy Fagherazzi[1]

[1]Department of Precision Health, Luxembourg Institute of Health, Strassen, Luxembourg
[2]Department of Infection & Immunity, Luxembourg Institute of Health, Esch-sur-Alzette, Luxembourg
[3]Department of Dermatology and Allergy Center, University of Southern, Odense, Denmark

**Correspondence to**
Dr Laurent Malisoux;
laurent.malisoux@lih.lu

## ABSTRACT

**Objective** To investigate if the physical activity (PA) prior to infection is associated with the severity of the disease in patients positively tested for COVID-19, as well as with the most common symptoms.

**Design** A cross-sectional study using baseline data from a prospective, hybrid cohort study (Predi-COVID) in Luxembourg. Data were collected from May 2020 to June 2021.

**Setting** Real-life setting (at home) and hospitalised patients.

**Participants** All volunteers aged >18 years with confirmed SARS-CoV-2 infection, as determined by reverse transcription-PCR, and having completed the PA questionnaire (n=452).

**Primary and secondary outcome measures** The primary outcome was disease severity (asymptomatic, mild illness and moderate illness). The secondary outcomes were self-reported symptoms.

**Results** From the 452 patients included, 216 (48%) were female, the median (IQR) age was 42 (31–51) years, 59 (13%) were classified as asymptomatic, 287 (63%) as mild illness and 106 (24%) as moderate illness. The most prevalent symptoms were fatigue (n=294; 65%), headache (n=281; 62%) and dry cough (n=241; 53%). After adjustment, the highest PA level was associated with a lower risk of moderate illness (OR 0.37; 95% CI 0.14 to 0.98, p=0.045), fatigue (OR 0.54; 95% CI 0.30 to 0.97, p=0.040), dry cough (OR 0.55; 95% CI 0.32 to 0.96, p=0.034) and chest pain (OR 0.32; 95% CI 0.14 to 0.77, p=0.010).

**Conclusions** PA before COVID-19 infection was associated with a reduced risk of moderate illness severity and a reduced risk of experiencing fatigue, dry cough and chest pain, suggesting that engaging in PA may be an effective approach to minimise the severity of COVID-19.

**Trial registration number** NCT04380987.

## Strengths and limitations of this study

► This is the first study to investigate the association between physical activity prior to infection and COVID-19 severity among people with mild and moderate courses in real-life settings.

► The study only includes adults with confirmed SARS-CoV-2 infection as determined by reverse transcription-PCR and classified as asymptomatic, mild or moderate cases according to an adapted version of the National Institute of Health symptom severity classification scheme.

► One of the main limitations of this study is that physical activity in the year before infection was assessed using a self-reported e-questionnaire, yet it covered all the physical activity domains (ie, occupational, transportation, leisure-time, household/gardening).

► Multinomial logistic regression models and separate logistic regression models were performed to investigate the association between physical activity and disease severity or specific symptoms.

► An in-depth analysis was conducted by controlling the models for the most relevant confounding factors identified so far.

to hospitals, intensive care units (ICUs) and outpatient facilities.[2] Epidemiological studies have demonstrated that mortality is higher among the elderly population, with a 6.1% increase in mortality per 10 years increase in age.[3] The risk for serious disease and death related to COVID-19 have been shown to be associated with baseline characteristics of patients such as old age, obesity, heavy smoking, as well as underlying conditions or comorbidities such as autoimmunity,[4] genetic errors of immunity,[5] hypertension, respiratory disease and cardiovascular disease.[6]

Physical activity (PA) is one of the leading determinants of health,[7] and thus, lack of PA may worsen the impact of the current pandemic. Indeed, the risk of developing

## INTRODUCTION

COVID-19, caused by SARS-CoV-2, spreads rapidly from China, caused outbreaks in countries throughout the world and was characterised by the WHO as a global pandemic on 11 March 2020.[1] This pandemic overwhelmed healthcare facilities, including but not limited

chronic diseases is much higher in those with low PA,[8 9] while COVID-19 patients with such underlying medical conditions (eg, obesity and diabetes) are more likely to be hospitalised and have a greater likelihood in poorer clinical outcomes.[10] It is also well established that insufficient levels of PA lead to reduced respiratory and cardiovascular capacities, which can lead to a greater occurrence of obesity and other chronic diseases.[11] Moreover, there is growing evidence that PA has a protective effect against infectivity and severity of respiratory infection due probably to a better immunological response.[12] Consequently, one may argue that both low PA, an important modifiable factor and high chronic disease prevalence worsen the severity of the crisis we are currently facing.

To date, the heterogeneity in the response to the infection to SARS-CoV-2 remains largely unexplained. COVID-19 symptoms are very heterogeneous and can range from minimal to significant severity in an infected individual.[13] A systematic review including 152 studies and 41 409 individuals showed that the most common symptoms were fever (59%), cough (55%), dyspnoea (31%), malaise (30%), fatigue (28%), sore throat (14%), headache (12%) and chest pain (11%).[14] While it has been demonstrated that PA decreases the risk of severe clinical COVID-19 outcomes (eg, hospitalisation or death),[15 16] there is still limited information on the impact of PA on the severity of COVID-19 in patients with less severe disease and on the risk of developing specific symptoms. Therefore, the primary objective of this study was to investigate if the level of PA over the year prior to infection is associated with the severity of the disease in patients positively tested for COVID-19. The secondary objective was to investigate if PA is associated with the most common symptoms: headache, sore throat, fever, dry cough, diarrhoea, breathing difficulties, loss of taste and smell, chest pain, muscle pain, fatigue, confusion and falls. We hypothesised that higher level of PA prior to infection would be associated with less severe forms of COVID-19, as well as with less frequent reports of the major COVID-19-related symptoms.

## METHODS
### Study design and participants
This is a cross-sectional study using data from a prospective, hybrid cohort study (Predi-COVID) composed of people positively tested for COVID-19 in Luxembourg.[2] The Predi-COVID study aims to identify epidemiological, clinical and sociodemographic characteristics as well as pathogen and/or host predictive biomarkers for the severity of COVID-19. The full study protocol has been published previously,[2] with some of the methods that are relevant to this study reproduced below. All volunteers received a full description of the protocol and provided written informed consent for participation. The findings from this study have been reported according to the Strengthening the Reporting of Observational Studies in Epidemiology statement.[17]

All individuals positively tested for COVID-19 in Luxembourg were eligible for the study and contacted by phone by the Health Inspection to enquire whether they consent to having their contact details communicated to the research team. The recruitment took place between May 2020 and June 2021. Inclusion criteria for this study were: having signed the informed consent, aged above 18 years, confirmed SARS-CoV-2 infection as determined by reverse transcription PCR (RT-PCR), performed by one of the certified laboratories in Luxembourg, and having completed the questionnaire on PA behaviour. Patients already included in another interventional study on COVID-19 and those unable to understand French or German were excluded from the study. The recruitment of participants depended on the emergence and spread of the virus and the resources available.

### Patient and public involvement
No patient or public involved.

### Outcomes
All clinical data were collected at baseline by research nurses using a modified version of the International Severe Acute Respiratory and Emerging Infection Consortium (ISARIC) case report form. The primary outcome was the severity of illness, which was classified using an adapted version of the National Institute of Health symptom severity classification scheme.[18] Participants were grouped into the following three categories: asymptomatic (positive RT-PCR test and no symptom), mild illness (positive RT-PCR test and one or more symptoms, but no shortness of breath, no symptoms of lower respiratory disease, no abnormal chest imaging) and moderate illness (positive RT-PCR test and symptoms of lower respiratory disease or abnormal chest imaging). The secondary outcomes were specific symptoms reported by the participants at baseline. The presence of the following twelve symptoms was considered for the present work: headache, sore throat, fever, dry cough, diarrhoea, breathing difficulties, loss of taste and smell, chest pain, muscle pain, fatigue, confusion and falls.

### Exposures
The exposure was PA over the year prior to infection, which was assessed using a self-reported e-questionnaire using the electronic patient-reported outcomes module of Ennov Clinical. The PA questionnaire included questions on weekly hours spent walking (to work, shopping and leisure time), cycling (to work, shopping and leisure time), gardening (and other handiwork), in household chores and sports activities (eg, racket sports, swimming, running) in the year prior to infection, each reported for winter and summer, separately.[19] The time reported for the two seasons was first averaged. Then each activity was assigned a metabolic equivalent task (MET) value based on the Compendium of PA,[20] which included MET values of 3.0 for walking and household, 4.0 for gardening and 6.0 for cycling and sports. A total weekly METs score

(in MET-hour/week) was then calculated from the self-reported data. In addition, PA was categorised into four according to METs score using quartiles.

## Covariates

Potential confounders were considered in the analyses and collected with the ISARIC case report form. They included age (years), sex, body mass index (BMI), as well as self-reported comorbidities, smoking status, income and sedentary behaviour. BMI was calculated as measured weight (kg)/height$^2$ (m$^2$). Comorbidities included hypertension, chronic heart disease, chronic pulmonary disease, asthma, chronic kidney disease, chronic kidney insufficiency with dialysis, liver disease (mild disease), liver disease (moderate or severe disease), chronic neurological disorders, malignant neoplasia/cancer, chronic haematological disease, AIDS, obesity, diabetes with complications, diabetes without complications, rheumatological disease, dementia, malnutrition and chronic obstructive pulmonary disease. As few participants experienced comorbidities, this variable was categorised into 'no comorbidity' and 'at least one comorbidity'. Participants were asked to report whether they are 'never smoker', 'former smoker' and 'current smoker'. Income was categorised into '<€3000/month', '€3000–€4999/month', '€5000–€10000/month' and '€>10000/month'. Sedentary behaviour was defined as self-reported average number of daily hours spent in sedentary behaviour (eg, at work, during meal, in front of the screen) prior to infection.

## Statistical analysis

Descriptive statistics of the study population are presented as counts and percentage for categorical variables and as median and IQR for not normally distributed continuous variables. Normality was assessed using Shapiro-Wilk test and histograms.

Multiple imputation was performed to deal with missing data. A multivariate imputation by chained equation (MICE) approach was used, assuming a missing at random mechanism. The best predictors were selected based on correlation with the outcomes[21] using the *quickpred* function from the *MICE* package in R. Ten datasets with 20 iterations were imputed and the plausibility of imputations were checked with density plots and summaries. Each imputed dataset was used separately to build the statistical models. Coefficients were pooled and confidence intervals were calculated based on Rubin's rules.[22]

Multinomial logistic regression models were used to investigate the association between PA and illness severity. Two different models were fitted: (1) unadjusted model (model 1), and (2) model 1 adjusted for age, sex, BMI, comorbidities, smoking status, income and sedentary behaviour (model 2). Separate logistic regression models (fully adjusted) were also used to investigate the association between PA and specific COVID-19 symptoms. For both outcomes, PA was considered as a continuous and a categorical variable in distinct models.

Cubic spline regression models were plotted to investigate the potential non-linear associations between PA and the risk of mild and moderate illness severity, compared with an asymptomatic form, as well as between PA and the risk of specific symptoms. Each cubic spline regression model was defined with four knots, placed at the tertiles of the PA distribution, and with a reference exposure value set at the median of PA for disease severity or a specific symptom, respectively. The *splines* R package was used to fit the models.

All the statistical analyses were performed in R (V.3.6.1) using RStudio (V.1.3.1093). Statistical significance was set to p<0.05.

## RESULTS

The analysis includes 452 adults, aged (IQR) 42 (31–51) years old, with confirmed SARS-CoV-2 infection who agreed to participate in the study and provided data on PA. Only five participants were hospitalised, but none of them was admitted to ICU. Thirteen per cent of the participants were asymptomatic (n=59), 63% were classified as mild illness (n=287), and 24% as moderate illness (n=106). The most prevalent symptoms were fatigue (n=294; 65%), headache (n=281; 62%), dry cough (n=241; 53%), muscle pain (n=237; 52%), sore throat (n=203; 45%), fever (n=197; 44%) and loss of taste and smell (n=179; 40%). Breathing difficulties (n=101; 22%), diarrhoea (n=89; 20%), chest pain (n=69; 15%), confusion (n=51; 10%) and falls (n=2; <1%) were less common.

Descriptive statistics of the study population stratified by illness severity are presented in table 1. Overall, the study population included 48% of women (n=216), median age was 42 (IQR: 31–51), BMI was 24.9 (IQR: 22.1–27.8) and 79% did not suffer from any comorbidity (n=359). Missing data varied from 0% to 5%. The variables that had missing data were income (n=21; 5%), sedentary behaviour (n=3; 0.66%), BMI (n=2; 0.44%), age (n=1; 0.22%) and smoking status (n=1; 0.22%).

Table 2 presents the unadjusted and adjusted models for the association between PA and disease severity. When PA was considered as a continuous variable, no association was found with mild or moderate forms of COVID-19 in the unadjusted model. After adjustment, greater PA was associated with a slightly lower risk of moderate illness (OR (95% CI): 0.99 (0.98 to 1.00), p=0.041). Cubic spline regression analysis showed that the relationship between PA and the risk of mild or moderate illness was not linear (figure 1), which supports the use of PA as a categorical variable. The unadjusted model did not reveal any association between PA categories and mild or moderate illness. However, the adjusted model showed a lower risk of moderate illness in the category with the highest PA level (OR (95% CI): 0.37 (0.14 to 0.98), p=0.045).

The associations between PA and specific symptoms in the adjusted models are presented in table 3. Greater PA was associated with lower risk of chest pain (OR (95% CI): 0.99 (0.98 to 1.00), p=0.007) when PA was considered

**Table 1** Descriptive statistics of the study population stratified by disease severity

| Characteristic | All (n=452) MED (IQR) or n (%) | Disease severity | | |
| --- | --- | --- | --- | --- |
| | | Asymptomatic (n=59) MED (IQR) or n (%) | Mild illness (n=287) MED (IQR) or n (%) | Moderate illness (n=106) MED (IQR) or n (%) |
| Age (years)* | 42 (31–51) | 43 (31–56) | 41 (31–51) | 42 (32–49) |
| Sex | | | | |
| Female | 216 (47.8) | 19 (32.2) | 134 (46.7) | 63 (59.4) |
| Male | 236 (52.2) | 40 (67.8) | 153 (53.3) | 43 (40.6) |
| BMI (kg/m²)* | 24.9 (22.1, 27.8) | 25.5 (22.2, 28.2) | 24.7 (22.1, 27.5) | 25.5 (22.2, 29.2) |
| Comorbidities | | | | |
| No comorbidities | 359 (79.4) | 42 (71.2) | 243 (84.7) | 74 (69.8) |
| At least one comorbidity | 93 (20.6) | 17 (28.8) | 44 (15.3) | 32 (30.2) |
| Smoking status* | | | | |
| Never smoker | 291 (64.4) | 34 (57.6) | 184 (64.1) | 73 (68.9) |
| Former smoker | 84 (18.6) | 13 (22.0) | 55 (19.2) | 16 (15.1) |
| Current smoker | 77 (17.0) | 12 (20.3) | 48 (16.7) | 17 (16.0) |
| Income (euro/month)* | | | | |
| <€3000 | 71 (15.7) | 11 (18.6) | 39 (13.6) | 21 (19.8) |
| €3000–€4999 | 110 (24.3) | 15 (25.4) | 70 (24.4) | 25 (23.6) |
| €5000–€10 000 | 203 (44.9) | 23 (39.0) | 138 (48.1) | 42 (39.6) |
| >€10 000 | 68 (15.0) | 11 (18.6) | 40 (13.9) | 18 (17.0) |
| Sedentary behaviour (hour/day)* | 7 (4, 10) | 6 (4, 10) | 7 (4, 10) | 6 (4, 9) |
| Physical activity (MET-hour/week) | 52.9 (30.8, 82.3) | 63.0 (40.3, 98.5) | 52.0 (31.4, 81.0) | 49.3 (27.4, 73.9) |
| Physical activity (MET-hour/week) | | | | |
| <30 | 108 (23.9) | 10 (16.9) | 68 (23.7) | 30 (28.3) |
| 30–52 | 113 (25.0) | 13 (22.0) | 75 (26.1) | 25 (23.6) |
| 52–82 | 116 (25.7) | 16 (27.1) | 74 (25.8) | 26 (24.5) |
| >82 | 115 (25.4) | 20 (33.9) | 70 (24.4) | 25 (23.6) |

*Imputed data were used for the descriptive statistics.
BMI, body mass index; MED, median; MET, metabolic equivalent task.

as a continuous variable. The category with the highest PA level was associated with lower risk of fatigue (OR (95% CI): 0.54 (0.30 to 0.97), p=0.040), dry cough (OR (95% CI): 0.55 (0.32 to 0.96), p=0.034) and chest pain (OR (95% CI): 0.32 (0.14 to 0.77), p=0.010). Figure 2 shows separate cubic splines investigating the association between PA and specific COVID-19 symptoms.

## DISCUSSION
The protective effect of meeting PA recommendations on the risk of severe COVID-19 outcomes (ie, death, ICU admission and hospitalisation) has previously been documented.[15 16] However, whether PA may also prevent less severe illness courses remains unknown. The primary objective of this study was to investigate if the level of PA over the year prior to infection is associated with the severity of the disease in patients positively tested for COVID-19. The secondary objective was to investigate if PA is associated with the most common symptoms such as fatigue, headache, dry cough, muscle pain, sore throat, fever and loss of taste and smell. We hypothesised that higher level of PA prior to the infection would be associated with lower risk of mild and moderate illness, and lower risk of suffering from some of the most commonly reported symptoms. Our main findings were that participants with greater PA were at a lower risk of moderate COVID-19 severity, which confirms our hypothesis. Furthermore, greater level of PA was also associated with a decreased risk of experiencing fatigue, dry cough and chest pain, which are among the most commonly reported symptoms in patients positively tested for COVID-19. These findings suggest that PA is a protective factor for

**Table 2** Associations between physical activity and illness severity

| Exposure | Outcome | Model 1 OR (95% CI) | P value | Model 2 OR (95% CI) | P value |
|---|---|---|---|---|---|
| PA (MET-hour/week) | Disease severity† | | | | |
| | Mild illness | 0.99 (0.99 to 1.00) | 0.106 | 0.99 (0.99 to 1.00) | 0.064 |
| | Moderate illness | 0.99 (0.99 to 1.00) | 0.068 | 0.99 (0.98 to 1.00) | 0.041* |
| PA (MET-hour/week)‡ | Disease severity† | | | | |
| 30–52 | Mild illness | 0.85 (0.35 to 2.06) | 0.717 | 0.75 (0.30 to 1.88) | 0.542 |
| 52–82 | Mild illness | 0.68 (0.29 to 1.60) | 0.378 | 0.55 (0.22 to 1.34) | 0.185 |
| >82 | Mild illness | 0.51 (0.22 to 1.18) | 0.117 | 0.46 (0.19 to 1.08) | 0.075 |
| 30–52 | Moderate illness | 0.64 (0.24 to 1.71) | 0.374 | 0.57 (0.20 to 1.58) | 0.278 |
| 52–82 | Moderate illness | 0.54 (0.21 to 1.40) | 0.205 | 0.48 (0.18 to 1.29) | 0.145 |
| >82 | Moderate illness | 0.42 (0.16 to 1.05) | 0.064 | 0.37 (0.14 to 0.98) | 0.045* |
| P-trend | Mild illness | 0.99 (0.98 to 1.01) | 0.374 | 0.99 (0.97 to 1.01) | 0.243 |
| P-trend | Moderate illness | 0.99 (0.97 to 1.01) | 0.203 | 0.99 (0.97 to 1.01) | 0.171 |

Values are presented as OR (95% CI), which were calculated according to Rubin's rule. All models were performed with imputed data. Model 1=unadjusted model; Model 2=model 1 adjusted for age, sex, BMI, comorbidities, smoking status, income and sedentary behaviour.
*P<0.05.
†Reference: asymptomatic
‡Reference: <30 MET-hour/week
BMI, body mass index; MET, metabolic equivalent task; PA, physical activity.

the development of moderate COVID-19 course in adults and for some common related symptoms.

Previous studies have shown that insufficient PA prior to the pandemic increased the risk of hospitalisation,[15 16 23–25] admission to ICU and death.[15 16] Notably, low PA was shown to be one of the stronger risk factors for severe COVID-19 outcome, after advanced age and history of organ transplant.[16] Furthermore, meeting the PA guidelines[26] has been shown to decrease the risk of SARS-CoV-2 infection in adult Koreans, beside the negative association with the risk of severe COVID-19 illness (ICU admission or administration of invasive ventilation) and COVID-19-related death.[15] Objective measures of PA have also demonstrated a decreased risk of contracting COVID-19 and hospitalisation in those with greater PA.[27] A study including only patients with chest CT scan confirming infection showed that physical inactivity

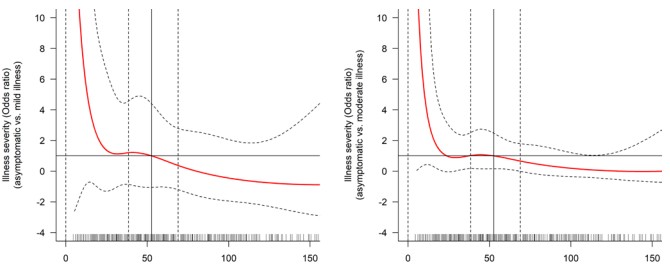

**Figure 1** Cubic spline regression investigating the association between physical activity and disease severity. Reference exposure value set at the median of physical activity (52.9 MET-hour/week). MET, metabolic equivalent task.

was associated with the severity of COVID-19 disease.[28] Overall, these previous studies suggested a protective effect of PA for severe COVID-19 outcomes, while some of these studies only included severe cases. While our findings are in line with these previous observations as they confirm the benefits of PA for COVID-19 severity, this is the first study to demonstrate that PA can also provide a protective effect for moderate courses.

Previous scientific literature has supported the role of PA against upper respiratory tract infections.[29] Research on the 2009 H1N1 influenza epidemic demonstrated a dose–response relationship between PA performed before infection and a reduction in the incidence, duration, or severity of acute upper respiratory tract infections.[30] During seasonal influenza, moderately active and active individuals were approximately 15% less likely to visit a physician or emergency services due to influenza compared with inactive individuals.[31] A recent meta-analysis revealed that people engaged in higher levels of PA showed a 31% risk reduction for community-acquired infectious disease.[12]

PA can play a protective role against respiratory viral infections and have important roles in a pandemic through three main mechanisms.

First, PA has an indirect protective effect by improving cardiovascular and respiratory functions (ie, the endurance and strength of the respiratory muscles) and lowering the risk of chronic diseases.[32] Consistently, maximal exercise capacity prior SARS-CoV-2 infection was shown to be inversely associated with the risk of hospitalisation due to COVID-19.[33] Exercise capacity is greatly influenced by PA, and more specifically regular moderate-intensity to

**Table 3** Associations between physical activity and specific COVID-19 symptoms using the adjusted model 2

| Exposure | Outcome | OR (95% CI) | P value |
|---|---|---|---|
| PA (MET-hour/week) | Symptom | | |
| | Fatigue | 1.00 (0.99 to 1.00) | 0.130 |
| | Headache | 1.00 (0.99 to 1.00) | 0.181 |
| | Muscle pain | 1.00 (0.99 to 1.00) | 0.442 |
| | Dry cough | 1.00 (0.99 to 1.00) | 0.056 |
| | Sore throat | 1.00 (1.00 to 1.00) | 0.973 |
| | Fever | 1.00 (0.99 to 1.00) | 0.453 |
| | Loss of taste and smell | 1.00 (0.99 to 1.00) | 0.286 |
| | Breathing difficulties | 1.00 (0.99 to 1.00) | 0.348 |
| | Diarrhoea | 1.00 (0.99 to 1.00) | 0.577 |
| | Chest pain | 0.99 (0.98 to 1.00) | 0.007* |
| | Confusion | 1.00 (0.99 to 1.01) | 0.804 |
| PA (MET-hour/week)† | Symptom | | |
| 30–52 | Fatigue | 0.63 (0.35 to 1.15) | 0.130 |
| 52–82 | | 0.69 (0.37 to 1.25) | 0.218 |
| >82 | | 0.54 (0.30 to 0.97) | 0.040* |
| 30–52 | Headache | 1.10 (0.62 to 1.94) | 0.745 |
| 52–82 | | 1.07 (0.61 to 1.89) | 0.807 |
| >82 | | 0.73 (0.42 to 1.26) | 0.256 |
| 30–52 | Muscle pain | 0.80 (0.46 to 1.39) | 0.430 |
| 52–82 | | 1.15 (0.67 to 1.99) | 0.615 |
| >82 | | 0.68 (0.40 to 1.18) | 0.171 |
| 30–52 | Dry cough | 0.66 (0.38 to 1.16) | 0.145 |
| 52–82 | | 0.69 (0.40 to 1.21) | 0.196 |
| >82 | | 0.55 (0.32 to 0.96) | 0.034* |
| 30–52 | Sore throat | 1.24 (0.71 to 2.16) | 0.453 |
| 52–82 | | 1.34 (0.77 to 2.33) | 0.297 |
| >82 | | 1.19 (0.69 to 2.07) | 0.531 |
| 30–52 | Fever | 0.97 (0.56 to 1.69) | 0.924 |
| 52–82 | | 0.90 (0.52 to 1.56) | 0.700 |
| >82 | | 0.89 (0.51 to 1.54) | 0.675 |
| 30–52 | Loss of taste and smell | 1.02 (0.58 to 1.79) | 0.947 |
| 52–82 | | 1.09 (0.62 to 1.90) | 0.769 |
| >82 | | 0.84 (0.47 to 1.48) | 0.539 |
| 30–52 | Breathing difficulties | 0.64 (0.33 to 1.25) | 0.190 |
| 52–82 | | 0.86 (0.45 to 1.62) | 0.634 |
| >82 | | 0.72 (0.38 to 1.37) | 0.318 |
| 30–52 | Diarrhoea | 0.72 (0.36 to 1.43) | 0.343 |
| 52–82 | | 0.94 (0.49 to 1.82) | 0.862 |
| >82 | | 0.70 (0.36 to 1.39) | 0.313 |
| 30–52 | Chest pain | 0.80 (0.39 to 1.65) | 0.553 |
| 52–82 | | 0.87 (0.43 to 1.78) | 0.705 |
| >82 | | 0.32 (0.14 to 0.77) | 0.010* |
| 30–52 | Confusion | 1.04 (0.40 to 2.69) | 0.941 |
| 52–82 | | 2.16 (0.91 to 5.09) | 0.079 |
| >82 | | 0.99 (0.38 to 2.61) | 0.982 |

Values are presented as OR (95% CI), which were calculated according to Rubin's rule. All models were performed with imputed data.
*P<0.05.
†Reference: <30 MET-hour/week.
MET, metabolic equivalent task; PA, physical activity.

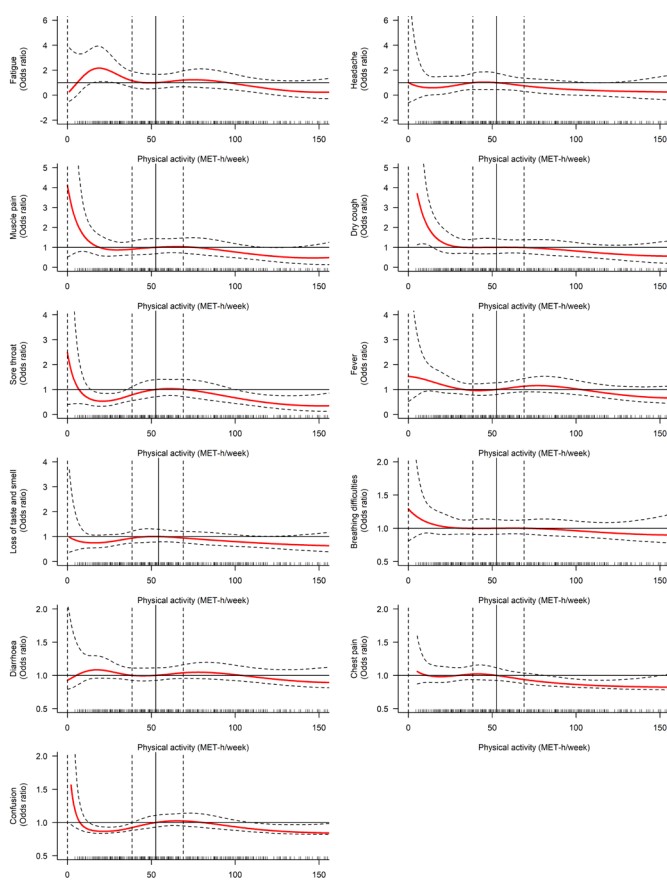

**Figure 2** Cubic spline regression investigating the association between physical activity and specific COVID-19 symptoms. reference exposure value set at the median of physical activity (52.9 MET-hour/week). MET, metabolic equivalent task.

vigorous-intensity aerobic exercise. The authors argued that exercise capacity is an important measure of overall health, the ability of the body to respond to external stressors, and more specifically, the ability to tolerate cardiopulmonary burden.[33]

Second, the immune system is very responsive to PA and exercise, with the extent and duration depending on the degree of physiological stress imposed by the workload. Importantly, most of the literature on the effect of PA on human immunity investigated acute effects of exercises and focused on athletes, which call for caution when generalising the findings. Globally, the beneficial effect of regular PA on the immune system may involve several mechanisms such as enhanced immunosurveillance, reduced systemic inflammation and improved regulation of the immune system as well as delayed onset of immunosenescence.[34 35] A recent systematic review investigated the effects of regular PA on the immune system.[12] Interventions including 3–5 sessions per week for an average of 30 min at moderate to vigorous intensity (eg, walking, running, cycling) resulted overall in a lower concentration of neutrophils, as well as a higher concentrations of CD4 T helper cells and salivary IgA. The lower concentration of neutrophil may be interpreted as a consequence

of the beneficial effect of regular PA on chronic inflammation.[36] CD4 T cells contribute to a rapid and more robust immune response. Salivary IgA can be regarded as the first line of defence of the immune system on the mucosal surface and plays other roles such as downregulating inflammation processes.[37] Among others, experimental studies have also showed that moderate intensity PA stimulates an increase in the antipathogen activity of immune system macrophages and anti-inflammatory cytokines in the blood, together resulting in a reduced influx of inflammatory cells into the lungs.[29]

Third, PA may also enhance vaccination response[38 39] and has a direct impact on trained immunity of innate immune cells such as Kupffer cells in the liver.[40] Trained immunity on the other hand, which describes a long-term boost through metabolic and epigenetic reprogramming of the innate immune response by certain stimuli (such as BCG vaccination or PA), has been proposed as an important tool for reducing susceptibility to and severity of COVID-19.[41]

Some studies have described a 'J' shaped association between exercise volume and infection with optimal protection at moderate levels of activity.[42] In this study, the cubic spline plots showed that the relationships between PA and COVID-19 severity and symptoms occurrence were not linear, but our sample size did not enable to define the shape of the curve accurately.

One of the limitations of this study was the use of self-reported measures to assess PA, which might have resulted in recall bias, compared with exposure assessment measured using objective means (ie, accelerometers), which can provide a more accurate assessment of the true level of PA. However, our PA assessment tool has previously been used in large cohort studies,[19 43] and covers all the PA domains (ie, occupational, transportation, leisure-time, household/gardening). Nevertheless, the use of self-report questionnaires usually leads to overestimation of PA, which may lead to underestimation of the magnitude of true associations.[44] Second, there was no measure of PA intensity, although each activity was assigned a specific MET value. Third, this study was an observational study with a limited sample size for some outcome categories. It is not possible to conclude that PA prior to infection is causally related to less severe COVID-19 outcomes as this study design suffers from a potential issue of residual confounding due to unmeasured or unknown confounders. However, our adjusted model controlled for all the most relevant confounders identified so far. Fourth, some estimated 95% CI suggest sparse data bias (see tables 2 and 3), which should be recognised as an important limitation.

## CONCLUSION
We found that greater PA prior to infection was associated with a reduced risk of moderate illness severity among adults positively tested for COVID-19. Greater PA was also associated with a reduced risk of experiencing

fatigue, dry cough and chest pain, which are among the most commonly reported symptoms in patients with COVID-19. This study provides new evidence that PA is a modifiable risk factor for COVID-19 severity, including moderate illness. Our findings suggest that engaging in regular PA may be one of the key actions individuals can take to minimise adverse consequences of COVID-19.

**Acknowledgements** We are thankful to the study participants, the Predi-COVID study group and the funders for their support of this initiative. The Predi-COVID study is supported by the Luxembourg National Research Fund (FNR) (Predi-COVID, grant number 14716273), the André Losch Foundation and by European Regional Development Fund (FEDER, convention 2018-04-026-21).

**Contributors** LM, AB, AF, GA, MO and GF contributed sufficiently to the manuscript to justify authorship. LM, AF, GA, MO and GF conceptualised the project, and LM, AB, GA and GF defined the methodology for the present study. LM and AB verified the underlying data and conducted the data analysis. All authors were involved in the interpretation of the analysis results. LM drafted the first manuscript and all other authors provided significant feedback and comments to refine the final manuscript. All authors approved the final manuscript and confirm that they accept responsibility to submit for publication. LM acts as a guarantor for the manuscript.

**Competing interests** None declared.

**Patient and public involvement** Patients and/or the public were not involved in the design, or conduct, or reporting, or dissemination plans of this research.

**Patient consent for publication** Consent obtained directly from patient(s)

**Ethics approval** The study was approved by the National Research Ethics Committee of Luxembourg (CNER) in April 2020 (ID: 202003/07).

**Provenance and peer review** Not commissioned; externally peer reviewed.

**Data availability statement** Data are available on reasonable request. As this is a cross-sectional analysis of baseline data from a currently ongoing prospective study, the data will not be made available publicly before the end of the Predi-Covid study. The study protocol can be found under https://bmjopen.bmj.com/content/10/11/e041834.abstract.

**ORCID iD**
Laurent Malisoux http://orcid.org/0000-0002-6601-5630

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
