## [Reviewer comments · BMJ Open]

ARTICLE DETAILS

TITLE (PROVISIONAL)	Associations between physical activity prior to infection and COVID-19 disease severity and symptoms: results from the prospective Predi-COVID cohort study
AUTHORS	Malisoux, Laurent; Backes, Anne; Fischer, Aurélie; Aguayo, Gloria; Ollert, Markus; Fagherazzi, Guy

VERSION 1 – REVIEW

REVIEWER	Nieman, David Appalachian State University, Health and Exercise Science
REVIEW RETURNED	14-Jan-2022

GENERAL COMMENTS	These data support a reduced risk of moderate COVID-19 for physically active individuals. This reviewer examined the paper closely. Although there were some limitations, these were adequately addressed in the discussion. The methods appear sound. The results are presented nicely. The introduction and discussion are well written. This reviewer has just one question and observation. The introduction states that the "primary objective of this study was to investigate if the level of PA prior to infection is associated with the severity of the disease in patients positively tested with COVID-19." But the discussion adds a different perspective claiming "the primary objective of this study was to investigate if PA is associated with illness severity among patients with asymptomatic, mild or moderate COVID-19 severity." This statement differs from the introduction, and a statement in the methods that claims "all individuals positively tested for COVID-19 in Luxembourg were eligible for the study..." This reviewer is left with the impression that individuals with severe COVID-19 did not agree to join the study. Or perhaps they were excluded. Please clarify this issue. This reviewer has no other concerns. The data are novel and interesting, and the paper is well written.
--

REVIEWER	Oliveira Souto, Fabrício Universidade Federal de Pernambuco, Nucleo de Ciencias da Vida
REVIEW RETURNED	15-Jan-2022

GENERAL COMMENTS	The authors investigated the relationship of physical activity before infection and disease severity in patients tested positively with COVID-19. In summary, it was observed that physical activity before COVID-19 infection presented a lower risk of moderate disease severity and a reduced risk of other symptoms, pointing to the practice of physical activity as a good approach to minimize the severity of COVID-19. .
--

	Some points about the work and suggestions: Line 64 and 65: The authors present a brief epidemiological overview, but without recent data on percentage, infection rate and mortality around the world. I suggest adding these elements. Line. 68 and 69: Epidemiological studies are again cited, but without presenting current numerical data to justify such statements. Line 92 e 93: The primary objective according to the present study seems to want to analyze the impact of physical activity before any infectious contact with COVID-19, however then the authors say that they used individuals who tested positive for the virus. The authors should clarify this objective statement better, after all, do the authors want to demonstrate that maintaining high levels of physical activity regularly is effective in mitigating symptoms during infection? Line 94: It would be interesting to describe all the symptoms. Line 136: We know the importance of using questionnaires in time of a pandemic by COVID-19. However, the following questions arise: is the questionnaire validated and reproducible for the public of the present study? Why were this questionnaire chosen, compared to tools for measuring physical activity levels, such as triaxial accelerometers? Line 138: Within the questionnaire were activities that require the subject to be away from home considered, could the data collection have been hampered by the lockdown period or the subjects' fear of going out on the streets? How do the authors deal with this situation and what are the impacts on their data? Line 140: Only at this moment that the justification for the primary objective appears, the authors will come to adjust. Line 164: Within the descriptive analysis, the authors do not report the use of a test to evaluate the distribution of data normality, why? Line 187: Was the sample calculated? When dealing with population-based studies, this is fundamental. The mean and standard deviation of the subjects' age should also appear in this part, I suggest including. In table 1, the present study uses participants with and without comorbidities, why? The presence of comorbidities impacts physical activity levels. Table 1 shows subjects with <30 METs, however, in table 2 this result does not appear, in fact I know that the objective of the present study is to observe groups with different levels of physical activity, mainly the active ones, however the data of the subjects in both the unadjusted and adjusted models would indeed demonstrate that in this study the reduction in physical activity was a risk factor, where we expected an odds ratio value >1. Why were these data not presented? Despite not being a direct measure of physical activity, the results of the present study are important for the scientific community. In the discussion, the authors initially deal with a summary of the main points and objectives of the study as well as the results found. The authors make a parallel with H1N1, most likely due to the scarcity of studies with COVID-19 and physical activity, in addition, when addressing the benefits in the infectious process, they cite physical exercise, which conceptually and functionally (systematized and with a well-defined objective) is different from physical activity. Line 252 to 264: The authors discuss little about the mechanisms that mediate between physical activity and the improvement in the health condition of these patients with COVID-19.
--	---

	Immunomodulation? Neuroendocrine stimuli? Increased physical fitness components? Which? Line 277 to 279: What physiological mechanisms related to physical activity mediate this improvement in cardiorespiratory fitness in COVID-19? Line 281 to 288: The immune system and its relationship with different manifestations of physical activity and exercise are approached superficially, only citing the benefits that occur, in fact I suggest that these benefits could be explained from a physiological point of view, which cells are activated? which pathways of signaling activities that lead to the promotion of these improvements? The discussion on which paths the improvement in symptoms that showed significance in the multivariate logistic regression occurs was not emphasized, only global benefits, why? This complicates the logical understanding between results and discussion, I suggest reformulating such questions and adding to the discussion connecting to the cited benefits. I acknowledge and thank Mr. Matheus Santos de Souza Fernandes for his support and additional comments for this review.
--	--

VERSION 1 – AUTHOR RESPONSE

Reviewer: 1 (Dr. David Nieman, Appalachian State University)

These data support a reduced risk of moderate COVID-19 for physically active individuals. This reviewer examined the paper closely. Although there were some limitations, these were adequately addressed in the discussion. The methods appear sound. The results are presented nicely. The introduction and discussion are well written. This reviewer has just one question and observation. The introduction states that the "primary objective of this study was to investigate if the level of PA prior to infection is associated with the severity of the disease in patients positively tested with COVID-19." But the discussion adds a different perspective claiming "the primary objective of this study was to investigate if PA is associated with illness severity among patients with asymptomatic, mild or moderate COVID-19 severity." This statement differs from the introduction, and a statement in the methods that claims "all individuals positively tested for COVID-19 in Luxembourg were eligible for the study..." This reviewer is left with the impression that individuals with severe COVID-19 did not agree to join the study. Or perhaps they were excluded. Please clarify this issue. This reviewer has no other concerns. The data are novel and interesting, and the paper is well written.

We would like to thank the reviewer for this positive feedback on our manuscript as well as for having pointed out this discrepancy in the description of the main objective in the introduction and discussion sections. We have aligned the text in the discussion, which now reads: "The primary objective of this study was to investigate if the level of PA over the year prior to infection is associated with the severity of the disease in participants positively tested for COVID-19." (p13, lines 246-247)

Indeed, all individuals positively tested for COVID-19 in Luxembourg were eligible for the study, and the physical activity questionnaire included questions on weekly hours spent walking, cycling, gardening, household chores, and sports activities in the year prior to infection. Therefore, the present study investigates the association between self-reported physical activity the year prior to infection and the severity of the disease in participants positively tested for COVID-19."

Overall, few participants with severe COVID-19 volunteered to participate in the Predi-Covid study. On top of that, none of them filled in the online questionnaire on physical activity. We would like to

emphasise that Predi-Covid was a prospective cohort study with a 1-year follow-up of the participants' health status and symptoms. It was expected that participants with a severe COVID-19 were not willing to volunteer for such type of study.

Reviewer: 2 (Dr. Fabrício Oliveira Souto, Universidade Federal de Pernambuco)

The authors investigated the relationship of physical activity before infection and disease severity in patients tested positively with COVID-19. In summary, it was observed that physical activity before COVID-19 infection presented a lower risk of moderate disease severity and a reduced risk of other symptoms, pointing to the practice of physical activity as a good approach to minimize the severity of COVID-19.

Thanks again for your constructive criticism of our manuscript. Please find below a point-by-point reply to your specific comments.

Some points about the work and suggestions:

Line 64 and 65: The authors present a brief epidemiological overview, but without recent data on percentage, infection rate and mortality around the world. I suggest adding these elements.

Thank you for this comment. The aim of the first sentence was not to present a brief epidemiologic overview, but the general context (i.e. the health condition investigated) with a reference to the origin of the pandemic. According to the suggestion of the reviewer to provide some numbers, we added the date when WHO made the assessment that COVID-19 can be characterized as a pandemic (i.e. on 11 March 2020; p4, line 66). Given the continuous evolution of the figures, we believe that it is not relevant to add information on infection rate and mortality around the world here.

Line. 68 and 69: Epidemiological studies are again cited, but without presenting current numerical data to justify such statements.

We agree with the reviewer that some figures are needed to support this statement. The sentence now reads: "Epidemiological studies have demonstrated that mortality is higher among the elderly population, with a 6.1% increase in mortality per 10 years increase in age." (p4, line 69)

Line 92 e 93: The primary objective according to the present study seems to want to analyse the impact of physical activity before any infectious contact with COVID-19, however then the authors say that they used individuals who tested positive for the virus. The authors should clarify this objective statement better, after all, do the authors want to demonstrate that maintaining high levels of physical activity regularly is effective in mitigating symptoms during infection?

Indeed, we investigated if the level of PA prior to infection is associated with the severity of the disease in participants positively tested with COVID-19. Individuals positively tested for COVID-19 in Luxembourg were invited to participate to the study. Then, they filled in a questionnaire on weekly hours spent walking, cycling, gardening, in household chores, and sports activities in the year prior to infection (each reported for winter and summer, separately). We did not assess trajectories, changes and regularity in physical activity. We clarified our aim, which now reads: "Therefore, the primary objective of this study was to investigate if the level of PA over the year prior to infection is associated with the severity of the disease in participants positively tested for COVID-19." (p5, line 93)

Line 94: It would be interesting to describe all the symptoms.

The list of the symptoms investigated in the present study was provided in the methods section of the initial version, lines 134-136. We have added the full list of symptoms at the end of the introduction of our revised manuscript according to the reviewer's request. (p5, lines 95-96)

Line 136: We know the importance of using questionnaires in time of a pandemic by COVID-19. However, the following questions arise: is the questionnaire validated and reproducible for the public

of the present study? Why were this questionnaire chosen, compared to tools for measuring physical activity levels, such as triaxial accelerometers?

Thank you for these relevant questions.

As we wished to gather information on the patient's physical activity over the year prior to infection and we recruited participants positively tested for COVID-19, we could not use a tool for objective measurement such as accelerometers, unfortunately. The data had to be collected retrospectively. We are aware of the limited validity and reproducibility of questionnaires on physical activity. The tool used in the present study was used in previous large cohorts (e.g., Fournier A, Dos Santos G, Guillas G, et al. Recent recreational physical activity and breast cancer risk in postmenopausal women in the E3N cohort. *Cancer Epidemiol Biomarkers Prev* 2014;23(9):1893-902; MacDonald CJ, Madika AL, Lajous M, et al. Associations between physical activity and incident hypertension across strata of body mass index: A prospective investigation in a large cohort of french women. *J Am Heart Assoc* 2020;9(23):e015121). Questions derived from the EPIC questionnaire, which has been validated against accelerometers (The InterAct Consortium. Validity of a short questionnaire to assess physical activity in 10 European countries. *Eur J Epidemiol* 2012; 27:15–25).

We first considered using the IPAQ, which is one of the most popular questionnaire. However, the IPAQ is meant to collect information on physical activity over the past week. As the physical activity questionnaire was filled in online by the participants after inclusion to the study, there was a risk that the past week corresponds to a period where the patient was already infected (which may have affected the level of physical activity).

Line 138: Within the questionnaire were activities that require the subject to be away from home considered, could the data collection have been hampered by the lockdown period or the subjects' fear of going out on the streets? How do the authors deal with this situation and what are the impacts on their data?

The questionnaire enquired on PA over the whole year prior to COVID-19 infection, so the data reflects a general pattern rather than a specific condition in the short-term. However, the reviewer is right that the one-year period may include some periods with strict lockdown, which may have influenced the level of physical activity of the patient. Still, our aim was to investigate the association between PA prior to infection and COVID-19 severity, whatever the determinants of that level of physical activity. The factors affecting the level of physical activity of the patient are independent of this association. The patient could have been less active over the previous year for many reasons (e.g. an injury, another health condition, lack of time for leisure-time PA, etc.).

Line 140: Only at this moment that the justification for the primary objective appears, the authors will come to adjust.

We understand the reviewer's concern about the consistency between the study objectives and the methods, especially the collection of physical activity data. The justification for the study objectives are presented in the introduction section and the main objective was clarified (p5, line 93). The information provided on line 140 is related to the methods used to collect physical activity data. We hope that our replies to your comments above and the change provide to the text are satisfactory and helped clarify a possible misunderstanding.

Line 164: Within the descriptive analysis, the authors do not report the use of a test to evaluate the distribution of data normality, why?

Thanks for pointing out this omission. We just forgot to add the information in the method section. Normality was assessed using Shapiro–Wilk test and histograms. The information has now been added on page 8, lines 168-169.

Line 187: Was the sample calculated? When dealing with population-based studies, this is fundamental. The mean and standard deviation of the subjects' age should also appear in this part, I suggest including.

No a priori sample size calculation was performed for the present study. As mentioned in the methods section, the present paper is an ancillary study using data from a prospective, hybrid cohort study: Predi-COVID. Predi-COVID was a study initiated very early in the pandemic, which aimed to understand the severity associated with a new, poorly understood pathogen. Therefore, the sample size was not formally determined. Recruitment of participants depended on the emergence and spread of the virus and the resources available. The latter information was added to the methods section. (page 6, lines 119-120)

We have followed the reviewer's suggestion and added median and interquartile range of the participants' age to the results section (page 8, line 191).

In table 1, the present study uses participants with and without comorbidities, why? The presence of comorbidities impacts physical activity levels. Table 1 shows subjects with <30 METs, however, in table 2 this result does not appear, in fact I know that the objective of the present study is to observe groups with different levels of physical activity, mainly the active ones, however the data of the subjects in both the unadjusted and adjusted models would indeed demonstrate that in this study the reduction in physical activity was a risk factor, where we expected an odds ratio value >1. Why were these data not presented? Despite not being a direct measure of physical activity, the results of the present study are important for the scientific community.

The reviewer is right, a substantial part of the population has comorbidities. These comorbidities could affect physical activity, similarly to sex, BMI, age and income. As these factors may also influence the severity of the disease, they were considered as potential confounders and we controlled for them in our adjusted models.

In table 2, the category "<30 METs" is not shown because this is the reference group (and therefore, the OR = 1).

In the discussion, the authors initially deal with a summary of the main points and objectives of the study as well as the results found. The authors make a parallel with H1N1, most likely due to the scarcity of studies with COVID-19 and physical activity, in addition, when addressing the benefits in the infectious process, they cite physical exercise, which conceptually and functionally (systematized and with a well-defined objective) is different from physical activity.

As mentioned by the reviewer, we made a parallel with studies on the protective effect of physical activity on H1N1 because of the limited data available on physical activity and COVID-19 when the present manuscript was submitted.

We also mentioned the benefits of physical exercise, as exercise is a form of physical activity, usually classified as vigorous physical activity. We think that this paragraph is relevant for the discussion as it presents a broader view on the relationship between physical activity (including vigorous intensity) and infectious diseases. Furthermore, the majority of the literature on the physiological mechanisms is devoted to the effect of vigorous intensity physical activity and exercises. Please, see our reply to your comment about "line 277-279".

Line 252 to 264: The authors discuss little about the mechanisms that mediate between physical activity and the improvement in the health condition of these patients with COVID-19.

Immunomodulation? Neuroendocrine stimuli? Increased physical fitness components? Which?

Thank you for your comment. We understand that the reviewer, and the readers, may expect that we elaborate on plausible physiological mechanisms that mediate between physical activity and COVID-19. We have provided additional information on these mechanisms in the paragraphs below (pages 14-15). The paragraph on line 252-264 addresses the relationship between physical activity and severe clinical COVID-19 outcomes such as hospitalisation, admission to intensive care unit and death. The take home message of this paragraph is that previous studies on physical activity only focused on severe clinical COVID-19 outcomes. As we only have few of them in our study (five participants were hospitalised, but none of them was admitted to ICU), we prefer elaborating on physiological explanation in the next paragraphs. Please, see below our replies to your comments.

Line 277 to 279: What physiological mechanisms related to physical activity mediate this improvement in cardiorespiratory fitness in COVID-19?

Thank you for your comment. The study we referred to in this paragraph revealed an independent and inverse association between maximal exercise capacity and the risk of hospitalisation due to COVID-19. However, the authors did not investigate the physiological mechanisms. We elaborate on the plausible explanations for this association in the discussion, as well as the connection with physical activity. The text reads as follow: “Consistently, maximal exercise capacity prior SARS-CoV-2 infection was shown to be inversely associated with the risk of hospitalization due to COVID-19.[33] Exercise capacity is greatly influenced by physical activity, and more specifically regular moderate- to vigorous-intensity aerobic exercise. The authors argued that exercise capacity is an important measure of overall health, the ability of the body to respond to external stressors, and more specifically, the ability to tolerate cardiopulmonary burden.[33].” (page 15, lines 283-287)

Line 281 to 288: The immune system and its relationship with different manifestations of physical activity and exercise are approached superficially, only citing the benefits that occur, in fact I suggest that these benefits could be explained from a physiological point of view, which cells are activated? which pathways of signalling activities that lead to the promotion of these improvements?

Thank you for your suggestion. We have reedited two paragraphs of our previous version and restructured the discussion as of line 279 in the new version of the manuscript. The paragraph addressing the responses of the immune system to physical activity and exercise has been developed to provide the readers with more thorough information on the pathways and mechanisms involved. The paragraph now reads as follow (p15, lines 288-304): “Second, the immune system is very responsive to PA and exercise, with the extent and duration depending on the degree of physiological stress imposed by the workload. Importantly, most of the literature on the effect of PA on human immunity investigated acute effects of exercises and focused on athletes, which call for caution when generalising the findings. Globally, the beneficial effect of regular PA on the immune system may involve several mechanisms such as enhanced immunosurveillance, reduced systemic inflammation and improved regulation of the immune system as well as delayed onset of immunosenescence.[34, 35] A recent systematic review investigated the effects of regular PA on the immune system.[12] Interventions including 3–5 sessions per week for an average of 30 min at moderate to vigorous intensity (e.g., walking, running, cycling) resulted overall in a lower concentration of neutrophil, as well as higher concentrations of CD4 T cell helpers and salivary immunoglobulin IgA. The lower concentration of neutrophil may be interpreted as a consequence of the beneficial effect of regular physical activity on chronic inflammation.[36] CD4 T cells contributes to a rapid and more robust immune response. Salivary IgA can be regarded as the first line of defence of the immune system and plays other roles such as down-regulating inflammation processes.[37] Among others, experimental studies have also showed that moderate intensity PA stimulates an increase in the antipathogen activity of immune system macrophages and anti-inflammatory cytokines in the blood, together resulting in a reduced influx of inflammatory cells into the lungs.[29].”

In addition, we have also included the aspect of “trained immunity” in the revised discussion, which describes the strengthening and metabolic as well as epigenetic reprogramming of innate immune cell responses through several health-promoting stimuli, including physical activity. The text reads as follow: “Third, PA may also enhance vaccination response [38, 39] and has a direct impact on trained immunity of innate immune cells such as Kupffer cells in the liver.[40] Trained immunity on the other hand, which describes a long-term boost through metabolic and epigenetic reprogramming of the innate immune response by certain stimuli (such as BCG vaccination or PA), has been proposed as an important tool for reducing susceptibility to and severity of COVID-19.[41]” (p15, lines 305-309)

The discussion on which paths the improvement in symptoms that showed significance in the multivariate logistic regression occurs was not emphasized, only global benefits, why? This

complicates the logical understanding between results and discussion, I suggest reformulating such questions and adding to the discussion connecting to the cited benefits.

The main purpose of the present study was to investigate the global health benefits (i.e. severity of the disease and presence of symptoms) of physical activity in the year prior to SARS-CoV 2 infection. Therefore, individuals positively tested for COVID-19 in Luxembourg were invited to participate to the study. Volunteers filled in a questionnaire on weekly hours spent in physical activities in the year prior to infection. They also reported their symptoms. The severity of illness was classified using an adapted version of the National Institute of Health symptom severity classification scheme, as described in the methods. Consistently, the major part of our discussion (interpretation of the results, comparison with the existing scientific literature, strengths and limitations) is related to this study purpose. We believe that our research question is properly formulated, and that the discussion is in line with the study purpose and findings. Nevertheless, we agree with the reviewer that readers may expect some developments on the plausible physiological mechanisms underlying the connection between physical activity and clinical COVID-19 outcomes. We hope that our answers to your comments above and the amendments provided to the manuscript are satisfactory (pages 14-15, lines 279-304).

I acknowledge and thank Mr. Matheus Santos de Souza Fernandes for his support and additional comments for this review.

We would like to thank Mr. Matheus Santos de Souza Fernandes for his contribution to the review of this manuscript.

VERSION 2 – REVIEW

REVIEWER	Oliveira Souto, Fabrício Universidade Federal de Pernambuco, Nucleo de Ciencias da Vida
REVIEW RETURNED	13-Mar-2022

GENERAL COMMENTS	Dr. Malisoux and collaborators presented a new and improved version of the manuscript. The authors demonstrated that physical activity before COVID-19 infection was associated with a reduced risk of moderate disease severity, suggesting that this practice may be an effective approach to minimizing the severity of COVID-19. Minor points: Please review parts of the text, such as the use of COVID-19 instead of Covid-19.
---